# Physiological and Medico-Social Research Trends of the Wave P_300_ and More Late Components of Visual Event-Related Potentials

**DOI:** 10.3390/brainsci11010125

**Published:** 2021-01-18

**Authors:** Sergey Lytaev, Irina Vatamaniuk

**Affiliations:** 1Lab of Applied Informatics, Lab of Autonomous Robotic Systems, St. Petersburg Federal Research Center of the Russian Academy of Sciences, 199178 Saint Petersburg, Russia; vatamaniuk.i.v@gmail.com; 2Department of Normal Physiology, Saint Petersburg State Pediatric Medical University, 194100 Saint Petersburg, Russia

**Keywords:** event-related potentials, visual evoked potentials, component P_300_, brain-computer interface, speller, oddball paradigm, categorization of images

## Abstract

To extend the application of the late waves of the event-related potentials (ERPs) to multiple modalities, devices and software the underlying physiological mechanisms and responses of the brain for a particular sensory system and mental function must be carefully examined. The objective of this study was aimed to study the sensory processes of the “human-computer interaction” model when classifying visual images with an incomplete set of signs based on the analysis of early, middle, late and slow ERPs components. 26 healthy subjects (men) aged 20–26 years were investigated. ERPs in 19 monopolar sites according to the 10/20 system were recorded. Discriminant and factor analyzes (BMDP Statistical Software) were applied. The component N_450_ is the most specialized indicator of the perception of unrecognizable (oddball) visual images. The amplitude of the ultra-late components N_750_ and N_900_ is also higher under conditions of presentation of the oddball image, regardless of the location of the registration points. In brain pathology along with the pronounced asymmetry of the wave distribution, reduction of the N_150_ wave and lengthening of its peak latency, a line of regularities were noted. These include–a pronounced reduction in peak latency P_250_ and N_350_, an increased amplitude of N_350_ in the frontal and central points of registration, a decrease in the amplitude of N_450_ in the left frontal cortex and its increase in the occipital registration points, activation of the occipital cortex at a time interval of 400–500 ms, as well as fusion later waves. We called such phenomena of the development of cognitive ERP in brain pathology “the incongruence of ERP components”. The results of the research are discussed in the light of the paradigm of the P_300_ wave application in brain-computer interface systems, as well as with the peculiarities in brain pathology.

## 1. Introduction

There is no doubt that the mechanisms of image recognition are of particular interest, since they characterize not only the final stage of perception, but are also an integral component or the initial stage of more complex mental functions–the level of wakefulness, attention, speech, thinking, emotions, motivation, as well as those based on their mental properties and states. For more than half a century, the parameters of the late positive oscillation of the brain evoked potentials, which has a latency period of about 300 ms and is called P_300_, have been used to assess the listed mental functions. The study of this component has evolved from a phenomenological description of its changes under various perceptual loads to the study of functional dependencies between physiological and mental content [1,2,3,4].

Since the first discovery of the P_300_, research has shown that this oscillation has two subcomponents. These are the new wave P3 (P3a) and the classic P_300_, which has since been renamed P3b. The P3a wave (new to P3) has a positive amplitude with a maximum deviation at the frontal/central registration points and a maximum peak latency in the range of 250–280 ms. The P3a wave has been associated with attention-seeking brain activity. In particular, it was registered during orientation, involuntary changes to the environment and processing of new information [5,6,7].

The P3b wave also has a positive deviation, appears a little later, after about 300 ms, and the peak changes at a latency of 250–500 ms or more, depending on the task. The largest amplitude, as a rule, was formed above the parietal registration points.

The P3b component has become an outstanding tool used to study cognitive processes, especially psychological research in the processing of information processes in the brain. It is generally accepted that the P3b wave is caused by unpredictable (low probability) events–the lower the probability of an event, the larger the P3b amplitude. However, in order to trigger P3b, the improbable event must be related in some way to the task at hand. For example, an incredible event might be an infrequent target letter in an email flow that a subject can respond to with a button click. Later, P3b was used to solve inverse problems–how difficult a task is for a cognitive workload [6,7]. 

Since the mid-1980s, one of the most talked applications about ERP components like the P_300_ has been in lie detection [8]. This practice has sometimes been used in some legal situations, while its use has declined in conventional polygraph research, in part due to unconscious and uncontrolled aspects of the P_300_. This method is based on the reproducible detection of the P_300_ wave, which is central to the idea of a multifaceted EEG response associated with memory and encoding (MERMER, Memory and Encoding Related Multifaceted Electroencephalographic Response) [9].

Three paradigms are currently used to generate P_300_ ERPs: single stimulus, oddball, and three stimuli. In each case, the subject is asked to monitor the achievement of the goal by pressing a button or mentally counting followed by a report. The single-stimulus paradigm irregularly presents only one type of stimulus, or a goal with zero appearance of any other type of target [10,11]. 

Applications have also been proposed for the interaction between the brain and the computer (Brain Computer Interaction, BCI). The P_300_ has a number of interesting qualities that aid in the implementation of such systems. First, the wave is constantly detected and triggered in response to precise stimuli. And secondly, the signal to register the P_300_ can be triggered in almost all subjects with slight differences in measurement methods, which can help simplify the interface design and improve usability. The speed at which the interface can operate depends on how much the detected signal is separated from the “noise”. One of the negative characteristics of the P_300_ is that the signal amplitude requires super-averaging several records over time to isolate the signal. This and other post-write processing steps determine the overall interface speed [12,13,14]. 

In the BCI system, the bioelectrical signal recorded from the scalp is considered as input. The digitized signal is sent to the processing layer, which extracts the necessary features through the application of spatial filtering and spectral analysis [15]. The extraction function is essential for encoding the intent or command of the user. To extract useful information, BCI can use the temporal parameters of the signal (peak latency of evoked potential, P_300_ components), frequency characteristics (EEG rhythms), spatial parameters, as well as a combination of all of these characteristics [13]. 

In general, modern BCI systems use a line of electrophysiological signals. Among them are visual and auditory evoked potentials, slow cortical potentials, alpha and beta EEG rhythm, and separately the P_300_ component of evoked potentials. The use of visual evoked potentials (VEP) as control signals requires muscular control of the gaze direction and, therefore, is BCI dependent. The other three electrophysiological signals are considered to be independent. 

To date, the main directions in the study of BCI systems have focused on two areas–visual evoked potentials, where the main indicator is the P_300_ wave with subsequent assessment of cognitive processes and EEG when performing or planning motor activity. 

Thus, using a publicly available database of 61-channel EEG signals recorded from 15 healthy persons during the performance of various movements of the right upper limb, a new classification system for the periods of pre-movement and rest, as well as pre-movement and pre-movement, has been developed [16]. For each time interval, the proposed system generates a time-frequency map of each signal source in the motor cortex using beam shaping and continuous wave conversion. Then all maps are embedded in the volume and used as input data for the Deep Convolutional Neural Network (CNN). The system was able to distinguish a preliminary movement from a state of rest with an average accuracy of 90.3% (min. 74.6%, max. 100%), surpassing comparable methods in the literature, and to distinguish a preliminary movement from a preliminary movement with an average accuracy of 62.47%. 

The bioelectrical activity of the brain is converted into an electrical signal during the signal acquisition phase [15]. Then the target user’s desire is extracted from the signal. For this, various electrophysiological characteristics can be used. Algorithms developed specifically for BCIs interpret the desired action and send it to an output device, which can be a display with letters or targets (P_300_ speller, alphabet), a wheelchair or neuroprosthesis (for example, a robot arm) [17]. Feedback helps maintain and improve the accuracy and speed of communication/action. Finally, the P_300_ component of the evoked potentials can be used as control signals for the BCI. In a number of studies, P_300_ BCI systems have been tested in people with disabilities [10,18].

In addition to the P_300_ component of visual evoked potentials, a BCI system is proposed for automatic control of the movement of wheelchairs and the detection of shadows along the route using decoded EEG signals from the user while driving the wheelchair [19]. The proposed algorithm significantly improves the efficiency and accuracy of shadow objects detection. Higher-order statistic functions such as skewness and kurtosis in addition to other optical characteristics are used as input to various machine learning classifiers. A Multilayer Perceptron, Autoencoder, 1D-CNN, and Support Vector Machine are used to accomplish the task of detecting shadow boundaries. The results showed that the proposed system outperforms all other modern approaches with an accuracy of 84.63%.

Concluding the introduction to the research part, it can be noted that with a variety of approaches in BCI systems, one of the leading problems is the low speed of information processing at the technical level. In principle, we cannot influence to the processing rate of human sensory information. However, the granularity of sensory and other cognitive processes can be explored in order to create new interfaces and software products.

The objective of this study was aimed to study the sensory processes of the “human-computer interaction” model when classifying visual images with an incomplete set of signs based on the analysis of early, middle, late and slow components of event-related potentials for healthy persons and brain pathology.

## 2. Materials and Methods

The studies on 26 healthy subjects (men) aged 20–26 years were performed. Event-related potentials were recorded using a computer neuromapper (Saico, Italy) in 19 monopolar sites from the surface of the head according to the 10/20 system with a reference electrode on the earlobe. The frequency bandwidth of the amplifier is 5–70 Hz. VEPs over 30 accumulations were averaged. Eye movements and other muscle artifacts under super-averaging conditions were assessed by recording purity and skin resistance in real time.

The investigated person was in a darkened screened room in a state of calm wakefulness, sitting in a comfortable chair. Stimulation was carried out by short-term (duration 0.3 ms, frequency 0.5 Hz) presentation of test images on the display screen, synchronized with the neuromapper computer. The distance from the observer to the screen is 2 m, the size of the images is 0.5 × 0.5 m.

The stimuli were two figures with an incomplete set of features (key and glasses). Our experience in testing both healthy and patients made it possible to use fragments of a key as a model of an identifiable image, the image of which usually does not cause difficulties (all 26 people were identified). On the contrary, the short-term presentation of image fragments of glasses, as a rule, is above the recognition threshold, even for healthy people with normal vision, which made it possible to use it as a model of an unrecognizable image (23 people were not identified). The image of key fragments was viewed as a simple cognitive stimulus. Fragments of glasses were considered as “oddball” visual image. 

Since the study of the main group was carried out in a clinical laboratory, registration of evoked potentials as diagnostic measures was also carried out in 97 patients of the clinic. These were patients with oncological (75 persons) and vascular pathology (24 persons) of the brain at the age from 29 to 68 years in a clear consciousness. The functions of visual perception corresponded to the age norm. All patients gave consent to research. In this article, we will focus only on some of the features of the registration of cognitive ERPs in brain pathology. The results of healthy subjects will be discussed in detail. 

ERPs were analyzed at a time interval of 1000.0 ms from the moment of stimulus delivery (Figure 1) using data from topographic brain mapping (in 19 sites), as well as an assessment of the spatio-temporal characteristics of evoked potentials (in 7 sites). The averaged ERP measured the amplitude (relative to the isoline) and peak latencies (PL) of all negative components (above the isoline) and one positive (below the isoline). The components have been designated N_70_, N_150_, P_250_, N_350_, N_450_, N_750_ and N_900_. Each peak of the VEP was identified automatically and visually, sometimes with a correction in unit ms. The calculation of the average amplitude of the peaks (from the isoline) at a fixed time interval: up to 400 ms–every 50 ms, then up to 1000 ms, every 100 ms was applied. 

The amplitude-time characteristics of evoked potentials were processed using two main types of multivariate analyzes–stepwise discriminant (to establish the main differences in amplitudes and peak latencies) and factorial (to assess the stability of these indicators) (BMDP Statistical Software). The final result for stepwise discriminant analysis is the F-criterion. Differences were considered significant at F > 4.0. Factor analysis was chosen as the second method for determining the stable and unstable characteristics of the evoked potentials, where the criterion for determining the serial number of a factor is the stability of the parameter. The most stable characteristics are shown in Figures 3–9. Such an approach in physiological studies is justified from the point of view that dispersion, being an indicator of stability, is a characteristic of the quality of regulation in biological systems and, especially, in brain structures [20]. At the start the data from all 19 leads were evaluated according to the 10–20 system. Subsequent analysis showed that to solve the tasks of establishing the role of the components, data on 7 points are sufficient. The discussion of the results is based on the presence or absence of differences, as well as the stability of amplitudes and peak latencies.

## 3. Results

Assessment of the amplitude-temporal characteristics of the VEP components and the data of topographic brain mapping during evoked potentials registration in response to stimulation by visual structures with an incomplete set of features shows a certain type of potential deployment in time (Figure 1 and Figure 2). These are early negative waves with PL of 50–100 ms (N_70_), intermediate components having a maximum amplitude at PL values of 120–200 ms (N_150_), as well as late negative fluctuations in the interval of the analyzed epoch 300–450 ms (N_350_ and N_450_). After that, a certain “lull” was noted in the VEP dynamics, corresponding to a time interval of 450–700 ms and representing the average EEG. Following this “quiet” area, EP oscillations were also noted, sometimes comparable in amplitude to the intermediate components and named by us N_750_ and N_900_. Among the positive waves, only an oscillation with a PL of 200–300 ms (P_250_) was analyzed, which, among other components located below the isoline had maximum amplitude and a stable manifestation.

Amplitude-time parameters of N_70_ (Figure 3) are characterized by stable manifestation only in the occipital points. In other parts of the cerebral cortex, they are less pronounced, both in amplitude and in functional stability, and in the frontal leads sometimes they are absent altogether, as can be seen in Figure 1 and Figure 2. At the same time, attention is drawn to the fact that the N_70_ amplitude in the central regions of the brain (F > 4.0) somewhat lightened during perception of a bizarre image. 

Brain mapping by component N_150_ (Figure 1, 105.0 ms; Figure 2, 130.0 ms) shows the maximum activation of the anterior brain areas, as well as some time epoch and a decrease in the amplitude (Figure 4) of this wave in the parietal and occipital sites. The amplitude characteristics of N_150_ are significantly higher in the central leads (C4–C3) during the perception of an identifiable object (F > 10.0), and the spatio-temporal parameters in this case have a stable manifestation. In the occipital and frontal areas of the cerebral cortex, the N_150_ amplitude is slightly different in both tasks (F < 4.0). At the same time, the presentation of a bizarre image is accompanied by more stable time indicators.

The P_250_ wave manifests itself (Figure 5), on the one hand, by a rather high stability of temporal parameters, especially in tests with stimulation by an identifiable object, and, on the other hand, by an almost complete absence of differences in amplitude characteristics (F < 4.0). Brain maps in both cases reflect the synchronous activation of all parts of the brain (Figure 1 and Figure 2).

According to our data, the most significant differences are noted after 250–300 ms from the moment of stimulus presentation. In terms of the degree of stability for all subsequent components of the evoked potentials (N_350_, N_450_, N_750_, N_900_), there is a high variability of peak latencies (low factor serial number) and, at the same time, stability of amplitude characteristics (maximum factor serial number). 

Thus, the amplitude of N_350_ (Figure 6) is significantly higher in the case of VEP registration in response to an identified signal. Attention is drawn to the fact of an increase in its amplitude from the occipital (O2–O1) to the central (C4–C3) and frontal leads (F4–F3). A similar tendency for the amplitude to increase from the occipital regions to the frontal-central regions is also noted in the dynamics of N_450_ (Figure 7). If in the occipital-parietal sites (O2, O1, Pz) the values of the N_450_ amplitudes in both samples practically do not differ (F < 4.0), then in the central (C4–C3) and frontal leads they significantly increase (F > 10.0) upon perception of “oddball” signal (Figure 1, 405 ms and Figure 2, 425 ms). 

The amplitude of the ultra-late components N_750_ (Figure 8) and N_900_ (Figure 9) is also higher under conditions of presentation of “oddball” image, regardless of the location of the registration points. It is noted that if, upon perception of an identifiable object, the amplitudes of N_750_ and N_900_ practically do not differ from each other in different leads or even approach the isoline (Figure 1), then in the case of stimulation by an unrecognizable object (Figure 2), the amplitude of N_750_ is facilitated from the back of the head to the forehead, and the dynamics of the N_900_ wears the opposite character (the amplitude is higher in the occipital sites). 

### Features of the Dynamics of Amplitude-Time Parameters of ERPs in Brain Pathology

In this article, we will present some general patterns of the dynamics of the amplitude-time parameters of ERPs in volumetric (oncology) and vascular pathological processes of the brain in comparison with healthy subjects, planning to devote a separate article to this topic in the future. 

Figure 10 presents a particular example of ERP registration in the perception of an identifiable image in a patient with meningioma of the left parietal lobe. There is a pronounced asymmetry in the distribution of waves with predominant activation on the affected side. If the N_150_ wave forms independently everywhere, the later waves in the 300–800 ms interval tend to merge without splitting into separate waves. 

This cerebral pathology in tests with the recognition and categorization of visual images is accompanied by a reduction in the N_150_ amplitude and an extension of its PL (except for frontal and brainstem pathology) in associative areas with predominance in the right frontal cortex. 

In case of cerebral pathology, the perception of the recognizable image is characterized by a pronounced reduction in the PL P_250_ and N_350_, while the perception of fragments of the “oddball” figure does not differ from the control parameters. In this case increased amplitude of N_350_ is recorded at the frontal and central registration sites during parietal, frontal and brainstem volumetric processes. Thus, it is possible that in case of brain damage, the recognition process in the positive variant ends earlier than in healthy subjects. In the opposite case, long-term memory is activated earlier (indicator–N_350_) at the junction of the associative and projection cortex. 

In left-sided ischemic stroke of the brain VEP is significantly reduced on the side of the stroke (Figure 11). If at the middle and right-hand points of registration all the components are well-formed in the recording interval of 1000 ms, on the left side all VEP waves are deformed and practically not generated. 

For the late ERP components, the nature of the adaptive mechanisms in cerebral pathology has a general trend with the control group, but with a more pronounced topographic dependence. Thus, in the first test, the N_450_ amplitude in the left frontal cortex decreases and increases at the occipital registration points in the absence of identification of the key fragments, as well as in the second test. Let us emphasize the activation of the occipital cortex at a time interval of 400–500 ms, which, in our opinion, is directly related to “extralogical” thinking. 

In right-sided meningioma ERP (Figure 12) in response to recognizable visual image a “destroyed” evoked potential is formed with an uneven manifestation of waves with a PL of 100–150 ms (N_150_). Late waves (300–800 ms) tend to either decrease significantly or merge. 

If in the first test sample in almost all categories of patients increased amplitude of N_750_ in the occipital cortex and N_900_ in the frontal cortex was recorded, then when the fragments of glasses are perceived, an increase in the amplitude of N_750_ is recorded only with damage to the temporal cortex. The N_900_’s relief in the second run is also more limited. 

## 4. Discussion

There are a line of indicators for assessing sensory processes. These include, in particular, the threshold of awareness of an external stimulus, characteristics of the orienting reflex, as well as bioelectric, autonomic, behavioral, emotional and other reactions [21]. It is believed that the greatest theoretical and methodological difficulties are caused by the indication of the awareness of the stimulus, for the study of which there are a number of specialized tests associated with the registration of electrographic characteristics in response to conscious and unconscious stimuli [8,22,23]. 

Discussion of the results obtained presupposes awareness of the presented stimuli followed by an invariant description by the subjects. The use of images with an incomplete set of features for these purposes is a convenient model for assessing the studied mechanisms of visual perception. Considering the perception of such images in healthy subjects and in brain pathology, we will attempt to analyze the role of the P_300_ wave and later ERP components as an indicator in BCI systems. 

So, the parameters of bioelectric activity in time sequence can be used in BCI systems. By origin, ERP components are divided into two types. Exogenous ERPs are the result of early automatic processing of the physical characteristics of external stimuli. Endogenous ERPs are the result of later cognitive signal processing. Conscious processing occurs with a delay of about 100–150 ms, when the visual signal is sent to the secondary projection zone of the visual system–the posterior parietal and inferior temporal cortex. Although the visual signal reaches the primary visual cortex 50–70 ms after the appearance of the stimulus, the behavioral response during this period is based on unconscious processing (reflexes) [2,6,24].

Traditionally, it is customary to associate the physiological genesis of early ERP waves (50–100 ms) with the primary cortical response reflecting the physical properties of the stimulus [25]. An analysis of the amplitude-temporal characteristics of N_70_, on the one side, confirms this position, since with the same energy intensity of both stimuli, the amplitude of N_70_ at the projection occipital registration sites slightly differs depending on the situational task. At the same time, its increase in the central brain regions under conditions of stimulation in a “eccentric” manner is characteristic, which can be regarded as the initial manifestation of the mechanisms of selective attention in the associative brain areas.

Figure 4 indicates the similarity of the processes that N_150_ reflects in its development, which is expressed in the temporary propagation of excitation from the projection to the associative cortex with the maximum manifestation in amplitude in the central areas. And the fact that the greatest degree of correlation between the evoked potentials configuration in different areas of the cerebral cortex is observed over a VEP time interval of 100–200 ms, which indicates optimal conditions for the exchange of neural information between these areas, is beyond doubt. However, attention is drawn to the fact that the absence of image recognition is accompanied by a significant suppression of the N150 amplitude, especially at the central registration points. On this basis, it can be assumed that the initial increased activation of selective attention mechanisms during the perception of a difficult-to-recognize object is subsequently replaced by a less pronounced excitation of the associative neural network during the exchange of incoming and stored information. 

The distribution of the P_250_ and N_350_ components, regardless of the perceptual task, simultaneously covers with approximately the same excitation, almost the entire cerebral areas (Figure 5 and Figure 6). The time interval for the formation of P_250_ is stable, in contrast to N_350_, which obviously indicates the same initial mechanisms for categorizing the image. At the same time, the facilitation of N_350_ in the task with the perception of an identifiable image, most likely, indicates the completion of the process of invariant recognition of the image, in contrast to the opposite task, accompanied by reduced amplitude of N_350_ with a subsequent increase in the amplitude of later components. 

Analysis of scientific literature sources shows that among the ERP components with a PL of more than 300 ms, negative waves in the 400–500 ms time interval and called either N_400_ or Nx-Ny are most studied [2,5,7,26]. And, if a wide variety of cognitive loads are associated with the ERP components with PL for about 300 ms, then N_400_ is considered a more specific response. The authors associate the increase in the N_400_ amplitude with the perception of new semantically significant information [8,27]. In some studies, an increase in the N_400_ amplitude was observed with an increase in the load on visual memory. In other works, the relief of N_400_ was noted upon the initial presentation of a picture or a word, and, conversely, the amplitude dropped upon repeated presentation of the object. On the basis of a series of experiments, where the evoked responses were compared when stimulated with “new” and “old” stimuli, a number of authors put forward a hypothesis that N_400_, in response to a new semantically significant stimulus, reflects the processes of an “exhaustive search” of all stimuli in memory with the aim of subsequent identification [11]. 

The results of this research show that a wave with PL 400–500 ms (N_450_) has a clear formation regardless of the perceptual task. The most pronounced amplitude is observed in the median and frontal leads when the “oddball” image is perceived. Similar data were also observed in studies with non-rhythmic presentation of words, where, in contrast to rhythmic presentation, the amplitude of N_450_ increased. Based on the foregoing, we can conclude about the specificity of negative waves with PL 400–500 ms reflecting the so-called “detailed process” or “exhaustive search” when identifying new information. 

In contrast to N_400_, the psychophysiological significance of the “late negative” waves (more than 600 ms) is less unambiguous. Some authors believe that waves with a PL of 600–800 ms characterize to a greater extent already known (repeated) images or faces [12,23]. In other studies, it is believed that the components of “late negative” ERP, as well as N_400_, reflect the processes of processing new unfamiliar information [28]. Of particular interest here are studies of the so-called reaction selection potential. In these experiments, the answer was absent for simple types of tasks presented (response time up to 450 ms) and, conversely, increased over a time interval of 600–800 ms with increasing test load complexity. 

The data of our research show that waves with PL of 700–800 ms (N_750_) have a more pronounced manifestation in response to unrecognizable images in the associative cortex. It can be assumed that, by analogy with the N_450_, the N_750_ component also reflects the “exhaustive search” processes for the subsequent categorization of the image. 

The appearance of VEP waves above 800 ms is also usually associated with the receipt of new information [7,21,23]. This was especially clearly demonstrated in a series of studies with sequentially presented stimuli [28]. In these experiments, as individual features of objects were presented, waves initially increased from PL 600–700 ms, and then in a time interval 800–1200 ms. In the present study, the N900 amplitude is more pronounced in response to “oddball” images, however, as compared to N_450_ and N_750_, the N_900_ amplitude has greater values in the occipital-parietal areas. 

Thus, all ERP components that form after 100 ms after the presentation of the visual image are considered as endogenous in origin with the inclusion of conscious processes, categorization, decision-making, as well as under the influence of emotions and motivations. In such a case, endogenous ERP components are appropriate for BCI systems as they are associated with subject recognition and attention. In addition to focus, fatigue and stressful conditions also affect the use of the BCI system [15]. Some patients with damaged prefrontal cortex associated with attention processes have atypical ERP formation [17]. As a consequence, there is an induction to use the BCI system [29]. 

It is the P_300_ wave that is the ERP that has attracted a lot of attention in BCI research [18,29]. Users (test subjects) can manipulate the amplitude of the P_300_, paying more attention to a specific event. Therefore, the P_300_ is used in many BCI systems to reveal intentions or information hidden in the EEG [30]. 

There is the following regularity–the less the probability of an event (identification, categorization, decision making), the greater the amplitude of P_300_. To induce a reliable P_300_ wave, the probability of the target stimulus is usually set at about 10% [6,7]. 

The BCI system based on the P_300_ has a very interesting and important property in that it does not require initial user training in order to register the P_300_ in response to the desired goal. Thus, the user can immediately start using it. This BCI property has been developed in systems for wheelchair control and neuroprostheses [17,29,31]. There are a number of prototypes for driving a wheelchair, however, movement is rather limited. A prototype of a brain-guided wheelchair has been developed that uses an EEG/P_300_ signal and a motion control strategy to navigate safely and efficiently in a building. With the P_300_ BCI system, the user can select a destination in the menu by counting the number of flashes of the destination [32]. 

An important application for the P_300_ BCI system is the spelling device [17], which allows patients with disabilities to communicate. BCI can provide an opportunity to answer simple “yes” or “no” questions at a speed of up to 20 bpm. In addition, BCI can provide slow text processing with a bit rate of about 2 wpm [15]. Despite the low transmission rate, it improves the quality of life of ALS patients. 

Thus, on the one side, the P_300_ is a typical answer to the task at hand, which is true for every person. However, on the other side, there are individual differences in the latency and amplitude of P_300_, which influence the interpretation of the results. 

If BCI systems based on the P_300_ are developed mainly for people with severe motor and sensory impairments, a prototype BCI for wheelchair control using decoded EEG signals from the user can serve as a certain point of contact. The prototype includes a shadow detection module based on an adaptive directional tracking filter to extract target features along the direction of the boundaries and offers a theoretical basis for wheelchair movement control [19].

Traditionally, the results of evoked potentials studies in brain pathology testify in favor of a reduction in the amplitude and lengthening of the latency of individual components [28]. However, as a rule, these studies are associated only with the assessment of the physical characteristics of the visual, auditory and somatosensory signals, which are reflected in the evoked potential up to 100–150 ms. Assessment of cognitive perception (more than 300 ms) according to ERP data in brain pathology is carried out less frequently. 

Along with the pronounced asymmetry of the wave distribution, reduction of the N_150_ wave and lengthening of its PL, a number of regularities were noted. These include–a pronounced reduction in PL P_250_ and N_350_, an increased amplitude of N_350_ in the frontal and central points of registration, a decrease in the amplitude of N_450_ in the left frontal cortex and its increase in the occipital registration points, activation of the occipital cortex at a time interval of 400–500 ms, as well as fusion later waves. We called such phenomena of the development of cognitive ERP in cerebral pathology the incongruence of ERP components. 

Similar patterns of restructuring of neural networks in the human brain are described when the threshold for image recognition is reached under conditions of uncertainty [33]. The results were registered using a combination of digital synthesis of dynamic images and the method of mapping brain activity according to BOLD (Blood-Oxygen-Level Dependent), functional magnetic resonance imaging signal. It has been shown that when the threshold for recognition of images of objects by a person is reached, the BOLD signal increases in some areas of the brain, and the activity of neural networks decreases in other areas of the brain. The brain zones are described, the activity of which is maximal during the time corresponding to the recognition threshold, the solution of the problem, and it is also shown that this activity is less under subthreshold and even suprathreshold conditions of observation of dynamic images.

## 5. Conclusions

1. A negative wave with a peak latency of 400–500 ms (N_450_) of ERP is the most specific indicator among all the components of the evoked response when perceiving “oddball” visual images.

2. The N_350_ component of ERP has large amplitude in the case of correct image recognition, which is regarded as the completion of the processes of invariant signal estimation at this stage. 

3. Late negative ERP waves with a peak latency of 600–1000 ms are more significantly pronounced in the perception of an unrecognizable image and are a reflection of the mechanisms of “exhaustive search” in order to categorize the incoming information. 

4. Components N_70_ and N_150_ ERP, respectively, reflect the initial and intermediate mechanisms of selective attention, which, in turn, depend on the final processes of perception. Completed invariant recognition is accompanied to a greater extent by N_150_ activation, while incomplete categorization is characterized by N_70_ relief followed by suppression of the N_150_ amplitude in the associative cortex.

5. Cognitive perception in brain pathology is characterized by incongruence of the visual event-related potentials, which is manifested by a reduction in the latent period of the formation of late waves, a significant increase in the amplitude of individual late waves in local brain areas, as well as the fusion of late components. 

The obtained and discussed results, mainly on healthy subjects, will serve as the basis for the subsequent analytical processing of studies in brain pathology. This article shows some general patterns of the dynamics of the amplitude-time parameters of ERPs in volumetric (oncology) and stroke pathological processes of the brain in comparison with healthy subjects, planning to carry out a detailed analytical research of the obtained results in the future. In addition, the present results can become a starting point for comparative video-flow and neurophysiological studies in assessing motor activity in various conditions.

## Figures and Tables

**Figure 1 brainsci-11-00125-f001:**
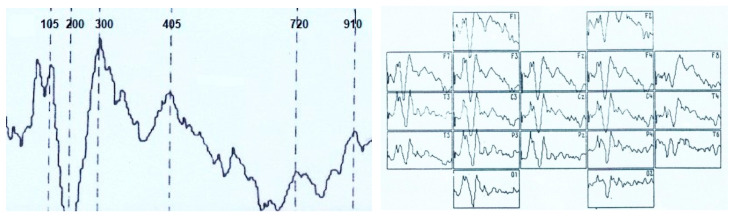
Visual evoked potential (numbers–time, ms) and VEP distribution according to the 10–20 system in the perception of an identifiable visual image. Analysis time—1000 ms.

**Figure 2 brainsci-11-00125-f002:**
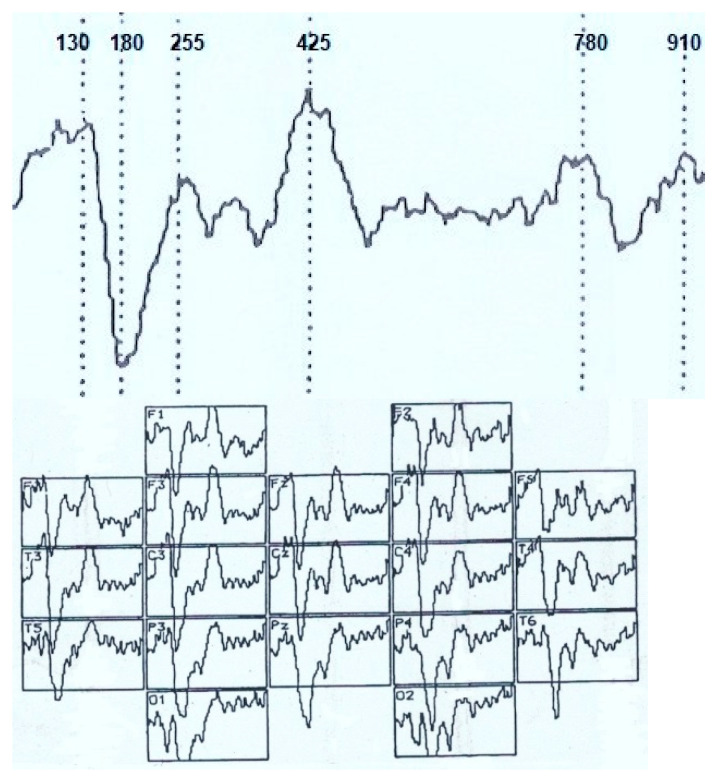
Visual evoked potential (numbers–time, ms) and distribution of VEP according to the 10–20 system during the perception of an unrecognizable (oddball) visual image. Analysis time—1000 ms.

**Figure 3 brainsci-11-00125-f003:**
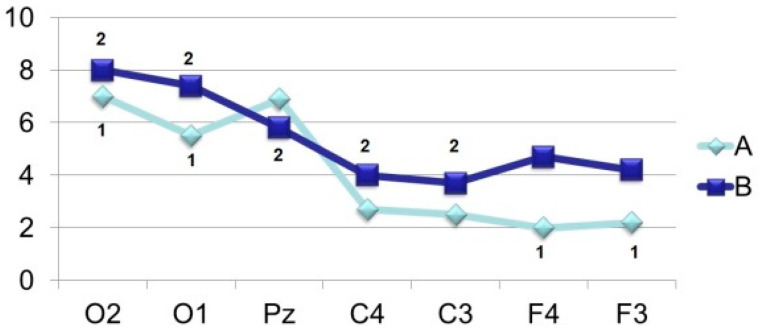
Averaged values of amplitudes and factor characteristics of amplitudes and PL of the N_70_ ERP component during the perception of recognizable (A) and unrecognizable (oddball) (B) images. On the abscissa–points according to the 10–20 system, on the ordinate–amplitude, uV. Figures on the graphs: 1–maximum factor number for amplitude, 2–maximum factor number for PL.

**Figure 4 brainsci-11-00125-f004:**
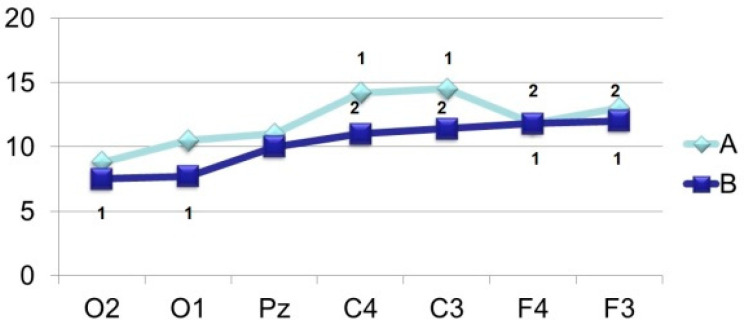
Averaged values of amplitudes and factor characteristics of amplitudes and PL of the N_150_ ERP component. Note. Legend in Figure 3.

**Figure 5 brainsci-11-00125-f005:**
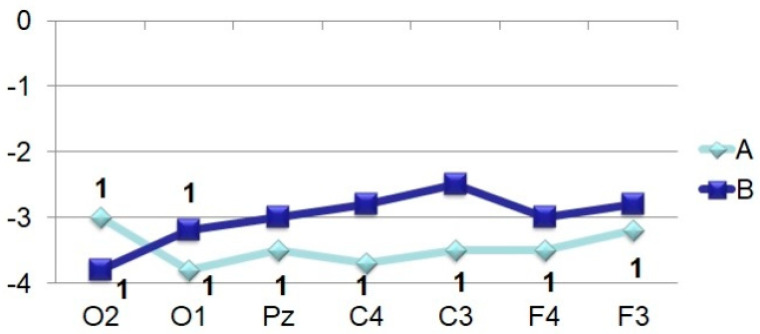
Averaged values of amplitudes and factor characteristics of amplitudes and PL of the N_250_ ERP component. Note. Legend in Figure 3.

**Figure 6 brainsci-11-00125-f006:**
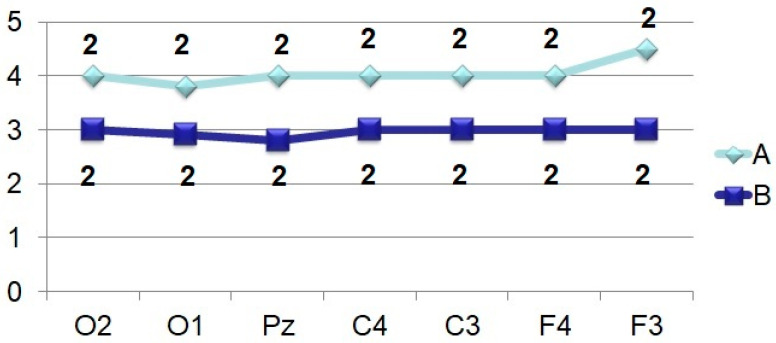
Averaged values of amplitudes and factor characteristics of amplitudes and PL of the N_350_ ERP component. Note. Legend in Figure 3.

**Figure 7 brainsci-11-00125-f007:**
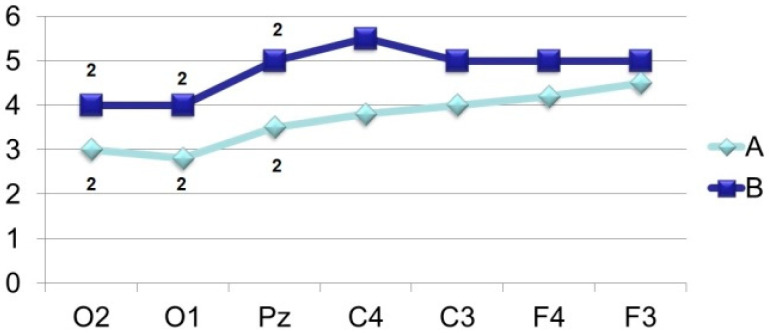
Averaged values of amplitudes and factor characteristics of amplitudes and PL of the N_450_ ERP component. Note. Legend in Figure 3.

**Figure 8 brainsci-11-00125-f008:**
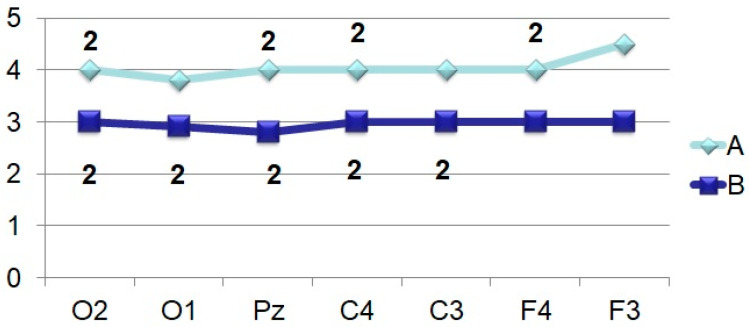
Averaged values of amplitudes and factor characteristics of amplitudes and PL of the N_750_ ERP component. Note. Legend in Figure 3.

**Figure 9 brainsci-11-00125-f009:**
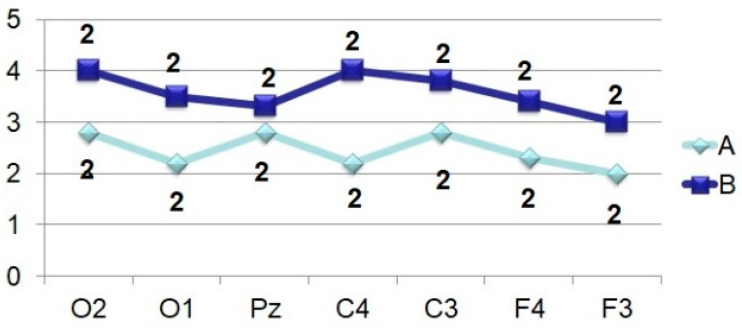
Averaged values of amplitudes and factor characteristics of amplitudes and PL of the N_900_ ERP component. Note. Legend in Figure 3.

**Figure 10 brainsci-11-00125-f010:**
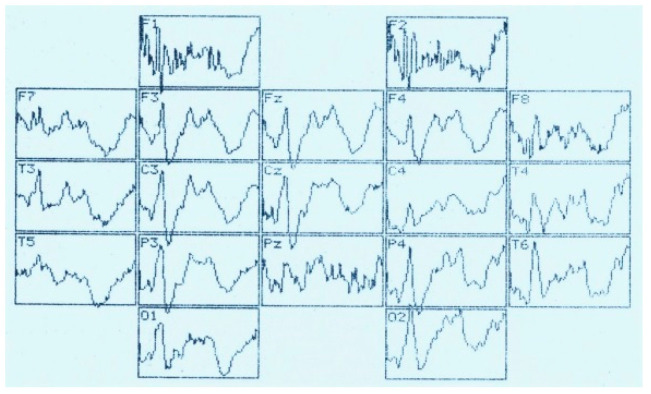
VEP distribution according to the 10–20 system in the perception of a recognizable visual image in a patient with meningioma of the left parietal lobe. Analysis time–1000 ms.

**Figure 11 brainsci-11-00125-f011:**
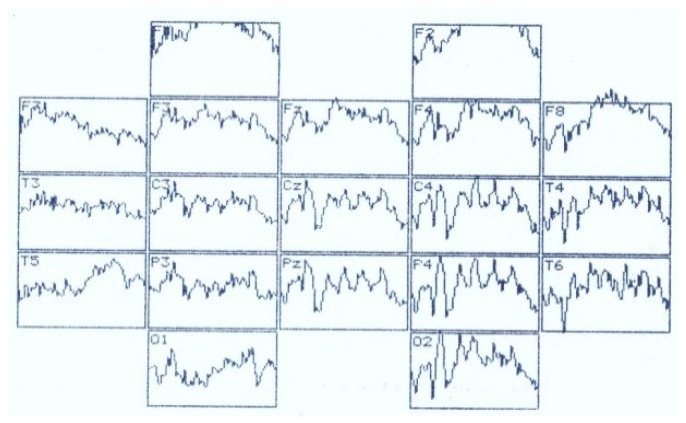
VEP distribution according to the 10–20 system in the perception of an identifiable visual image in a patient with left-sided ischemic stroke. Analysis time—1000 ms.

**Figure 12 brainsci-11-00125-f012:**
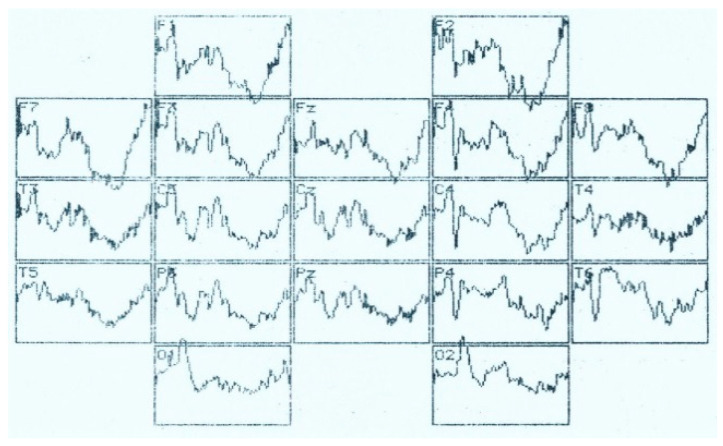
Distribution of VEP according to the 10–20 system in the perception of a recognizable visual image in a patient with meningioma of the right parietal lobe. Analysis time–1000 ms.

## Data Availability

Not applicable.

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
