# Peer review of "Physiological and Medico-Social Research Trends of the Wave P300 and More Late Components of Visual Event-Related Potentials"

_brainsci, 2021, doi:10.3390/brainsci11010125_

Round 1

Reviewer 1 Report

The objective was aimed to study the sensory processes of the “human-computer interaction” model when classifying visual images with an incomplete set of signs based on the analysis of early, middle, late and slow components of event-related potentials (ERPs) (See lines 13-15). In general, the article has been well written. The aim study is also well founded. The interesting experimental design has been proposed. The results are interesting for a developing of new approaches to BCI. However, data processing, statistical analysis and results are not described in sufficient details. For this reason, the results look doubtful. I recommend adding new details for data processing ans statisitcal analyses so that the results can become reproducible.

My critical comments:

1) It was not specified which software was used for EEG recording, stimuli presentation, calculation of VEPs’ amplitudes, and statistical processing of data?

2) I recommend adding a figure with cortical topography of amplitude for several local peak of VEPs. 

3) It was not mentioned, how the eyes' blinking and other muscle’s artefacts were rejected?

4) Which approach was applied for identification of local VEPs peaks? Each peak was identified by visual inspection of time-amplitude plots or automatically?

5) Which approach was chosen for calculation of amplitude for each VEP peak? There are two possible ways: 1) calculation of the mean amplitude at a fixed time-interval (if it is so, you need to specify time-windows for each peak in the methods); 2) calculation of the maximum positive amplitude for a positive peak (for example for P300) and the maximum negative amplitude for negative peaks (if it is so, you need to specify how the amplitude maximum was determined for each peak).

6) The design of statistical analysis of results should be essentially revised. One-way ANOVA has not been clearly described. I did not understand, which indeed analysis was performed for selection of the most informative indicators (lines 127-130). The effect of which factor was estimated? Was it performed separately for each of EEG channels? If yes, how the topographical properties of VEP were analyzed? I didn’t see any information about a correction for multiply comparison, but without such correction the statistical significance of results looks dubious.

7) In line 130-131. Statistical approach for the second case is not presented at all.   

8) In the results' session, only the Fisher coefficient value (F) was specified, but that is not enough. You have to also specify degrees of freedom, significance level (p), and effect size (ƞ2) for each result. Without such parameters, your results do not look rigor. (I recommend to present the effects of ANOVA in such form: F(1, 25) = 3,48; p = 0,035, ƞ2 = 0,34.)

Author Response

Dear Reviewer!

We greatly appreciate your work in reviewing our paper. We understand your remarks, and we tried to take them into account almost completely in the new version of the article. All changes in this version are highlighted in yellow.

  1. Software, recording, etc. Some indexed journals have lately strongly opposed the names of commercial products, considering it as advertising. Therefore, in the first version, we reflected this information rather succinctly. Here we have added information about the neuromapper (Saico, Italy) and the soft - BMDP (c 2017 nQuery) (19, 135, 173).

  1. There was a fairly homogeneous group of healthy subjects. The cortical topography as a whole is shown in Fig. 1 and 2. We plan to analyze these studies in more detail in brain pathology in the future, where local scalp peaks are more interesting.

  1. Eye movements and other muscle artifacts are important factors in neurophysiology. My clinical experience suggests that these factors are more important for spontaneous EEG. For evoked potentials under super averaging conditions, these factors are less significant. The main thing is the cleanliness of the recording. Either there is cleanliness or not. We proceeded from this (137-139).

  1. Each peak was identified both automatically and visually, sometimes with a correction in unit ms (168-170).

  1. Used the calculation of the average amplitude of the peaks (from the isoline) at a fixed time interval. Intervals added. (168-170)

6, 7, 8. Since we did not specify the software product (BMDP) and its features in the first version, such questions were formed. We used two main types of multivariate statistics analysis – stepwise discriminant (to establish the main differences in amplitudes and peak latencies) and factorial (to assess the stability of these indicators). Yes, these kinds of multivariate statistics in the soft use dispersion characteristics. But the final result for the stepwise discriminant analysis is the F value. For the factor analysis, it is classified into factors. Stepwise discriminant analysis in clinical neurophysiology is quite common. Factorial is used less often and was chosen as the second method for determining stable and unstable EP characteristics (171-184).

Hope we took into account your comments as much as possible.

Authors

Reviewer 2 Report

The paper is interesting and reports promising results. However, I have some comments:

  • results need to be reported in the abstract
  • main contributions need to be listed in the introduction
  • Not clear the novelty of the paper. The methodology needs to be further detailed and discussed
  • Conclusion needs to be revised and future works need to be reported 
  • Recent advances in BCI need be included in the paper. In order to give the reader an up-to-date picture the authors can refer to: i) A deep CNN approach to decode motor preparation of upper limbs from time–frequency maps of EEG signals at source level; ii) A Novel Approach to Shadow Boundary Detection Based on an Adaptive Direction-Tracking Filter for Brain-Machine Interface Applications
  • Quality of Figures from 3 to 9 need to be significantly improved

Author Response

Dear Reviewer!

We greatly appreciate your work in reviewing our paper. We understand your remarks, and we have tried to take them into account almost completely in the new version of the article. All changes in this version are highlighted in yellow.

The most important results and statements have been added to the annotations (12-14, 22-29).

A number of methodology changes have been made (168-184).

Initially, we did not set the task in the introduction to make a comprehensive overview of the research of the BCI system. We have concentrated only on evoked potentials. That being said, we are familiar with deep learning research (CNN) and have added recommended papers to the introduction (88-105), discussion, and bibliography (ref. 16, 19).

In the conclusion we have added our ideas about future. First of all, we are talking about the analytical processing of research in brain pathology. And subsequently - comparative video streaming and neurophysiological studies in assessing motor activity in various conditions.

Figures 3-9. In the new version we increased the size of the pictures. Changing the design is very easy and takes a few seconds. But practice shows that when it comes to multichannel comparative data of evoked potentials, this format is quite clear for neurophysiologists. Amplitude in microvolts, sites 10-20 and factor characteristics of stable parameters provide enough information.

Hope we took into account your comments as much as possible.

Authors

Round 2

Reviewer 2 Report

The comments have been addressed accordingly. The paper can be now accepted for publication.